# Synthesis of Superionic Conductive Li_1+x+y_Al_x_Si_y_Ti_2−x_P_3−y_O_12_ Solid Electrolytes

**DOI:** 10.3390/nano12071158

**Published:** 2022-03-31

**Authors:** Hyeonwoo Jeong, Dan Na, Jiyeon Baek, Sanggil Kim, Suresh Mamidi, Cheul-Ro Lee, Hyung-Kee Seo, Inseok Seo

**Affiliations:** 1School of Advanced Materials Engineering, Jeonbuk National University, Baekje-daero 567, Jeonju 54896, Korea; grassmarket@naver.com (H.J.); ld3310@jbnu.ac.kr (D.N.); asssa2089@naver.com (J.B.); agaleon@daum.net (S.K.); sureshmamidi@jbnu.ac.kr (S.M.); crlee7@jbnu.ac.kr (C.-R.L.); 2Future Energy Convergence Core Center, School of Chemical Engineering, Jeonbuk National University, Baekje-daero 567, Jeonju 54896, Korea; hkseo@jbnu.ac.kr

**Keywords:** Ionic conductivity, LASTP, all-solid-state battery, relative density, activation energy

## Abstract

Commercial lithium-ion batteries using liquid electrolytes are still a safety hazard due to their poor chemical stability and other severe problems, such as electrolyte leakage and low thermal stability. To mitigate these critical issues, solid electrolytes are introduced. However, solid electrolytes have low ionic conductivity and inferior power density. This study reports the optimization of the synthesis of sodium superionic conductor-type Li_1.5_Al_0.3_Si_0.2_Ti_1.7_P_2.8_O_12_ (LASTP) solid electrolyte. The as-prepared powder was calcined at 650 °C, 700 °C, 750 °C, and 800 °C to optimize the synthesis conditions and yield high-quality LASTP powders. Later, LASTP was sintered at 950 °C, 1000 °C, 1050 °C, and 1100 °C to study the dependence of the relative density and ionic conductivity on the sintering temperature. Morphological changes were analyzed using field-emission scanning electron microscopy (FE-SEM), and structural changes were characterized using X-ray diffraction (XRD). Further, the ionic conductivity was measured using electrochemical impedance spectroscopy (EIS). Sintering at 1050 °C resulted in a high relative density and the highest ionic conductivity (9.455 × 10^−4^ S cm^−1^). These findings corroborate with the activation energies that are calculated using the Arrhenius plot. Therefore, the as-synthesized superionic LASTP solid electrolytes can be used to design high-performance and safe all-solid-state batteries.

## 1. Introduction

In recent years, renewable energy generation and storage have garnered attention for alleviating emerging environmental concerns. Renewable resources intermittently produce energy [1,2] and require energy storage devices to maintain continuity. Li-ion batteries (LIBs), which have moderate energy and power densities, are currently used in a wide range of applications [3,4]. However, LIBs that use liquid electrolytes suffer from safety issues, such as explosions or fires [5,6]. Some studies report on non-flammable liquid electrolytes, but they require an additional battery pack to secure sealing and avoid leakage [7,8,9]. Research on solid electrolytes has been conducted to design safe, leakage-free solid-state batteries that use Li metal as the anode and have a theoretical specific capacity of 3860 mAhg^−1^, enabling high-energy-density batteries [10,11,12].

Electrolytes for all-solid-state batteries are divided into oxide-based, sulfide-based electrolytes and polymer electrolytes [13,14,15,16,17]. Sulfide-based electrolytes have relatively high ionic conductivities compared to oxide electrolytes but react with moisture in the air to generate hydrogen sulfide, which is a toxic substance [18]. On the other hand, oxide-based electrolytes have higher chemical stability than sulfide-based electrolytes. Perovskite, garnet, and sodium superionic conductors (NASICON) are commonly used oxide-based electrolytes [19]. Among these oxides, NASICON has high ionic conductivity and superior H_2_O stability [20,21,22]. However, the low ionic conductivity of solid electrolytes compared to that of commercial liquid electrolytes hinders their practical application.

LiM_2_(PO_4_)_3_ (M = Ti, Zr, Ge) and related materials exhibit a NASICON structure and are widely used in the synthesis of solid electrolytes for solid-state batteries [23]. NASICON structures are rhombohedral (space group R3$\bar{C}$) [24,25] and consist of a TiO_6_ octahedron and PO_4_ tetrahedron. According to their lattice structure, the migration of Li ions is driven by thermally activated hopping through available sites [26]. The mobility of Li ions can be affected by the partial substitution of Ti^4+^ with other atoms, such as Ge, Al, Hf, Sn, Zr, and Si, and thereby leads to changes in the crystal structure, which are mainly determined by the size of the LiO_6_ octahedron. If Ti^4+^ is replaced by smaller ions, the space available for the LiO_6_ octahedron is sufficiently large, and the activation energy for Li-ion migration decreases as Li prefers to move toward a larger polyhedron [27,28,29].

This study aims to optimize the synthesis conditions to fabricate highly ionic conductive solid electrolytes. A novel NASICON-type Li_1.5_Al_0.3_Si_0.2_Ti_1.7_P_2.8_O_12_ (LASTP) solid electrolyte was synthesized using a solution-based method. The as-prepared powder was calcinated at 650 °C, 700 °C, 750 °C, and 800 °C to determine the optimum synthesis conditions for high-quality LASTP. Later, the LASTP solid electrolytes were sintered at 950 °C, 1000 °C, 1050 °C, and 1100 °C to determine the optimum relative density and ionic conductivity. The ionic conductivity of the LASTP solid electrolyte was measured using electrochemical impedance spectroscopy (EIS) and correlated with the calculated activation energy from the Arrhenius plot. This work demonstrates that superionic solid electrolytes can be successfully synthesized as potential candidates for all-solid-state batteries.

## 2. Materials and Methods

### 2.1. LASTP Powder Synthesis

A solution-based method was used for the synthesis of the LASTP powder. Stoichiometric amounts of NH_4_H_2_PO_4_ (98.0%, Samchun, Seoul, Korea), LiCl (98.2%, Samchun, Seoul, Korea), Al(NO_3_)_3_ 9H_2_O (98.0%, Samchun, Seoul, Korea), Ti(C_4_H_9_O)_4_ (98.0%, Samchun, Seoul, Korea), and Si(C_2_H_5_O)_4_ (98.0%, Aldrich, St. Louis, MO, USA) were dissolved in deionized (DI) water and stirred for 2 h until the solution turned into a gel-like precipitate. The mixture was ball-milled using a planetary mill with Zr balls at 450 rpm for 1 h. Thereafter, the solution was stirred again for 30 min for uniform mixing. The precipitate was dried in an oven at 80 °C for 12 h. As-prepared LASTP powder was placed in an alumina crucible and calcined at various temperatures ranging from 650 °C to 800 °C in the air atmosphere. The temperature was increased at a ramp rate of 2 °C/min till the desired temperature and then maintained constant for 12 h to complete the calcination. Finally, the samples were cooled to room temperature in the furnace for 10 h. The calcined powders were ground using an agate mortar and labeled LASTP-X (where X is the respective calcination temperature).

### 2.2. Fabrication of the LASTP Pellet

The synthesized LASTP-X powder (5 g) was added to DI water (4 mL) in a planetary mill and rotated at 300 rpm for 30 min. Polyvinyl alcohol (5 wt%) was added as a binder and mixed uniformly at 450 rpm for 20 min. A fixed quantity of powder was compressed at 60 MPa for 20 s to prepare pellets. As-prepared pellets were sintered at various temperatures ranging from 950 °C to 1100 °C in the air atmosphere. Initially, the pellets were heated to the desired temperature at a ramp rate of 2 °C/min and dwelled for 6 h. Later, the samples were cooled to room temperature in the furnace for 12 h.

### 2.3. Material Characterizations

The surface morphologies of the LASTP-X powders and pellets were examined using field-emission scanning electron microscopy (FE-SEM; SU-70, Hitachi, Tokyo, Japan) at an accelerating voltage of 10 kV. Energy-dispersive X-ray spectroscopy (EDXS) was performed to determine the elemental distribution of LASTP.

The structural properties of the LASTP powders and pellets were analyzed using X-ray diffraction (XRD; D/MAX-2500, Rigaku, Tokyo, Japan) with a Cu-K radiation source. XRD patterns were obtained over the 2θ range of 10–70° at a step size of 0.02° and a scan rate of 2°/min.

### 2.4. Electrochemical Test

The electrochemical properties of the LASTP pellets were evaluated using a Won-A Tech EIS tester. Both sides of LASTP-X pellets were sputtered with Cu to measure the resistance using EIS (0.1 Hz–1 MHz). The activation energy was calculated from EIS data collected over a temperature range of 25 °C to 100 °C at a step size of 25 °C.

## 3. Results and Discussion

From FE-SEM images, the LASTP-X powder synthesized at different calcination temperatures (X = 650 °C, 700 °C, 750 °C, and 800 °C) showed highly interconnected particles ranging from 150 to 350 nm in size (Figure 1). The particle size increased slightly with the calcination temperature owing to the enhanced crystal growth kinetics at elevated temperatures. However, the non-uniform size of the particles may adversely affect Li-ion migration pathways; therefore, the sintering process is required to maintain a uniform particle shape and size.

XRD was performed to confirm the purity of the synthesized LASTP-X crystals. Figure 2a compares the XRD patterns of the prepared powder with those of standard LiTi_2_(PO_4_)_3_ (LTP), (JCPDS #35-0754), and the peaks were identified and indexed. The stoichiometry composition of the LASTP is Li_1.5_Al_0.3_Si_0.2_Ti_1.7_P_2.8_O_12_. Since there was no standard JCPDS or ICDD data available for LASTP powders, the objective of preparing the NASICON-type LASTP crystal is achieved by comparing the as prepared LASTP powder patterns with standard NASICON type LTP patterns. The 20–30° (2θ) range was magnified to perform a detailed peak analysis, as shown in Figure 2b. In the XRD patterns of LASTP-650 and LASTP-800, the relative peak intensity at 24.6° did not match the standard, and secondary phase peaks emerged at approximately 28° and 29.5°. Thus, from the comparative XRD structural analysis, the LASTP powders calcined at 750 °C have the best match with the standard pattern, whereas the other samples have shown the presence of secondary phases (AlPO_4_, SiO_2_, TiP_2_O_7_) as shown in Figure 2b. Therefore, the calcination temperature of 750 °C was set for the further processes.

EDS results of the LASTP-750 powder are shown in Figure 3b–g. Figure 3a shows an image of the randomly selected particles. No unwanted elements were found in the elemental composition of the LASTP-750 powder (Figure 3b). The elemental EDS mapping in Figure 3c–g confirmed that Al, Ti, P, O, and Si were uniformly distributed in the LASTP-750 sample. The Li atom distribution was beyond the instrument’s capability and could not be shown. 

FE-SEM cross-sectional images of pellets of LASTP-750 sintered at various temperatures (950 °C, 1000 °C, 1050 °C, and 1100 °C) are shown in Figure 4. Particle agglomeration occurred during the sintering of the LASTP pellets, resulting in increased grain size and density, as shown in Figure 4a–d. The particle and grain sizes also increased with temperature. However, the pellets sintered at 1100 °C (Figure 4d) had more space owing to the increased irregular grain size, which decreased the density. Several spaces and cracks are observed owing to the stresses caused by anisotropic thermal expansion in LASTP [30,31]. For LASTP-750 sintered at 1100 °C, Li-ion mobility may be restricted by the large grain boundaries. Therefore, sintering under optimized conditions enables the appropriate particle shape, size, and density for facilitating enhanced Li-ion diffusion. 

The dependence of the relative density on the sintering temperature was examined. The relative density was expressed as the ratio of the pellet density to the theoretical density. Figure 5 and Table 1 show the relative densities of the LASTP-750 pellets sintered at 950 °C, 1000 °C, 1050 °C, and 1100 °C. The relative densities of the LASTP-750 pellets were low when sintered at 950 °C (63.30%) and 1000 °C (67.23%) owing to less aggregation at low sintering temperatures. The pellet sintered at 1050 °C exhibited the highest relative density (96.84%), while the pellet sintered at 1100 °C fractured due to thermal shock.

Figure 6a–c depict the Nyquist plots for each calcination temperature for samples sintered at 950 °C, 1000 °C, and 1050 °C, respectively. The Nyquist plots consisted of a semicircle at high frequencies and a straight line with a slope at low frequencies. The impedance spectrum was composed of the bulk resistance (R_b_), grain boundary resistance (R_gb_), and Warburg impedance (Z_w_). The ionic conductivities of the pellets were calculated at various calcination and sintering temperatures and were compared. Figure 6d shows the temperature dependence of the ionic conductivity, where the ionic conductivity increased with the sintering temperature. However, the ionic conductivity of the pellets sintered at 1100 °C could not be measured owing to material fracture. The measured pellet thickness and ionic conductivity values under all synthesis conditions are listed in Table 2. The pellets calcined at 750 °C and sintered at 1050 °C exhibited the highest ionic conductivity. 

Figure 7 shows the impedance spectra of LASTP-750 pellets at different sintering temperatures and the equivalent circuit diagram with related equations. R_b_, R_gb_, CPE, and Z_w_ represent the bulk resistance, grain boundary resistance, constant-phase element, and Warburg impedance. The Nyquist plot was fitted to the equivalent circuit as shown in the inset of Figure 7. 

The resistance of the pellets decreased with the increasing sintering temperature. The bulk and grain boundary ionic conductivities of these pellets are listed in Table 3. As the sintering temperature increased, the grain boundary-ionic conductivity increased more than the bulk ionic conductivity. The increase in the grain boundary-ionic conductivity enhanced the total ionic conductivity, which is the sum of the bulk and grain boundary-ionic conductivities [32]. The pellet sintered at 1050 °C exhibited the highest ionic conductivity (9.455 × 10^−4^ S cm^−1^) owing to the high relative density of the pellet, which may improve the Li-ion mobility in the electrolyte.

The activation energy (*E_a_*) was calculated from the Arrhenius plot of the ionic conductivity. Arrhenius plots of pellets sintered at 950 °C, 1000 °C, and 1050 °C are shown in Figure 8. The *E_a_* for different sintering temperatures was calculated using the following expression: (1)$$\sigma_{T}T=Aexp\left(-E_{a}/kT\right)$$
where *A* is the pre-exponential factor; *k* is the Boltzmann constant; *T* is the absolute temperature, and *σ_T_* is the total ionic conductivity. EIS data were collected from 25 °C to 100 °C with a step size of 25 °C to calculate the activation energies of the pellets. As presented in Table 4, *E_a_* ranged between 0.226 and 0.263 eV. The activation energy is the energy barrier related to the transport of Li ions through the crystalline structure of LASTP [33]. Comparison of ionic conductivities and activation energies of the various solid electrolytes are tabulated in the Appendix A. The lowest *E_a_* (0.226 eV) was achieved for the pellet sintered at 1050 °C, which also demonstrated the highest ionic conductivity. Therefore, superionic solid electrolytes can be successfully prepared and applied in high-performance and safe all-solid-state batteries.

## 4. Conclusions

This study demonstrates the optimized synthesis conditions of high ionic conductive LASTP solid electrolytes. In the optimization, it was observed that as the sintering temperature increased, the relative density increased with the pellet agglomeration, leading to a synthesis of high ionic conductive LASTP. The ionic conductivity of the pellet sintered at 1050 °C was found to be 9.455 × 10^−4^ S cm^−1^. Moreover, the activation energy studies were well corroborated with the ionic conductivity and revealed that the solid electrolyte sintered at 1050 °C has the lowest *E_a_* value of 0.226 eV. Therefore, this study provides an efficient method for preparing superionic solid electrolytes which can be deployed in all solid-state batteries to enable a safe and high-performance battery operation.

## Figures and Tables

**Figure 1 nanomaterials-12-01158-f001:**
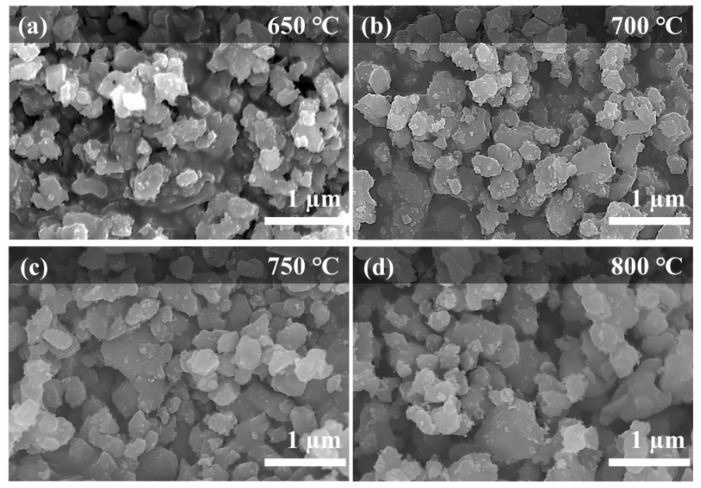
FE-SEM images of the LASTP powder calcined at various temperatures (**a**) 650 °C, (**b**) 700 °C, (**c**) 750 °C, and (**d**) 800 °C.

**Figure 2 nanomaterials-12-01158-f002:**
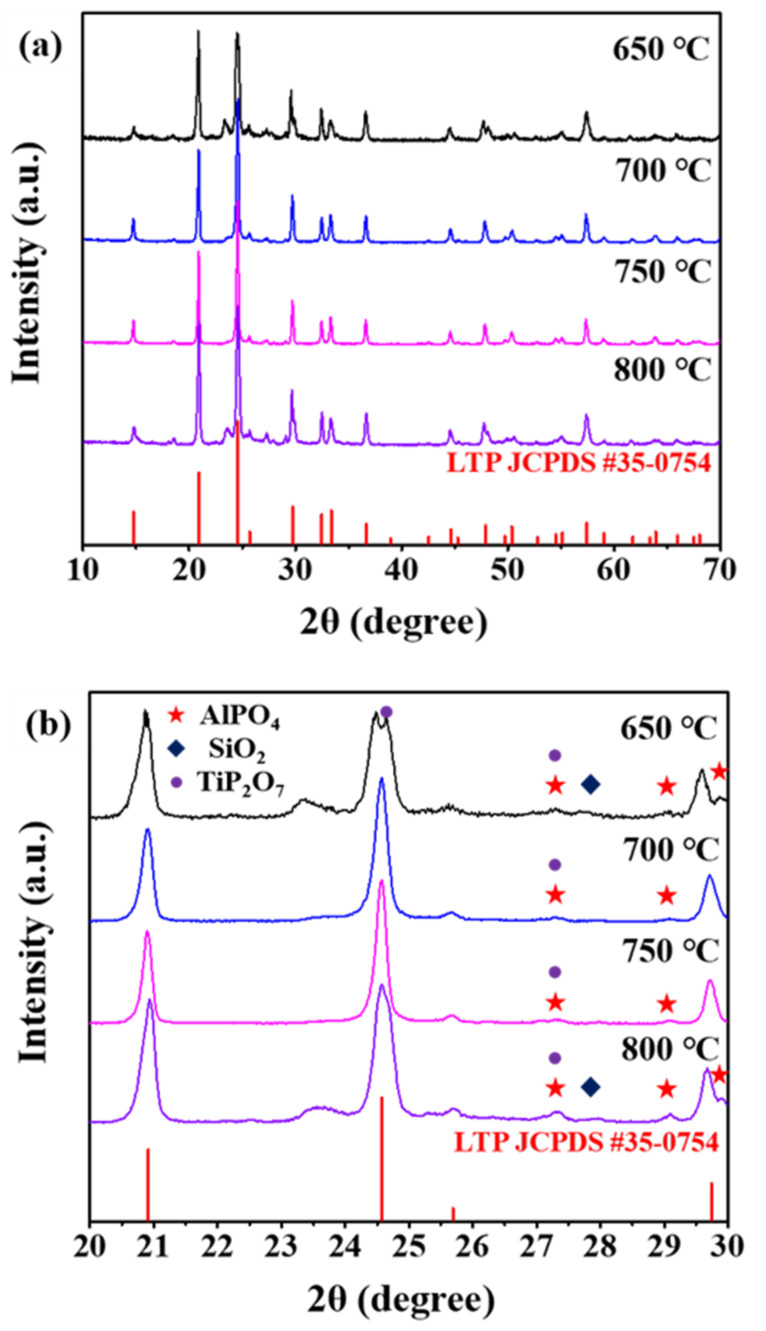
XRD patterns of (**a**) the LASTP-X powder calcined at various temperatures along with standard LTP JCPDS card and (**b**) rocking curves from 20° to 30°.

**Figure 3 nanomaterials-12-01158-f003:**
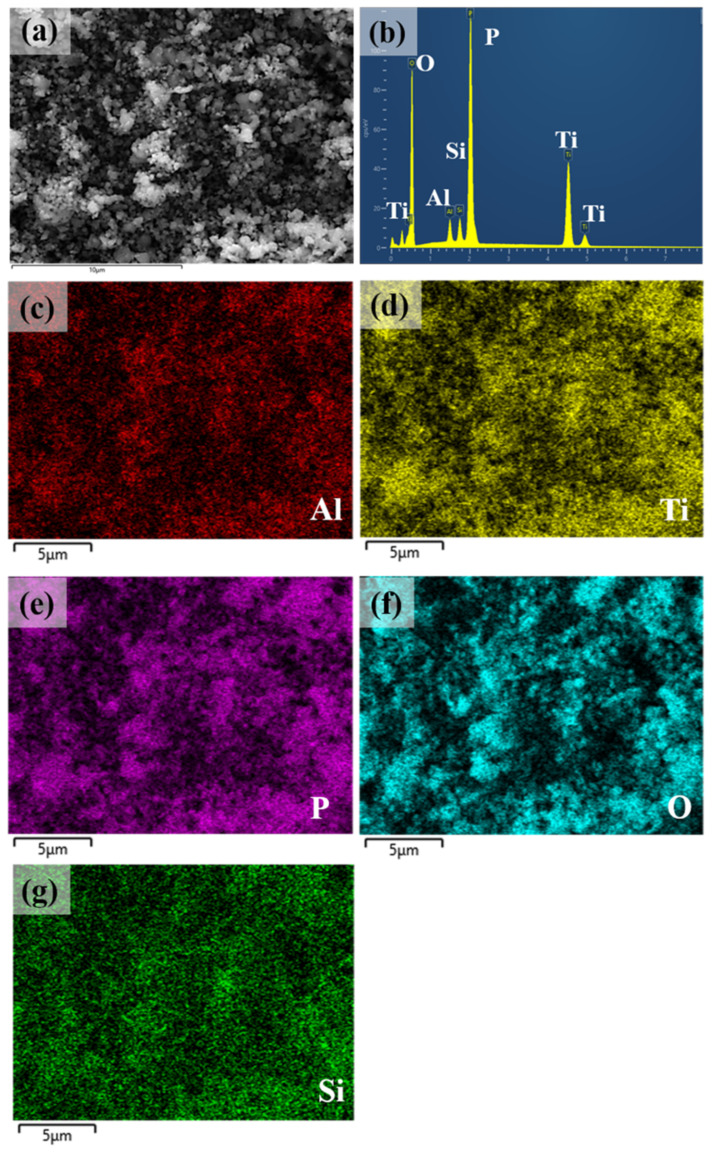
(**a**) SEM image of LASTP-X powder, (**b**) EDS spectra of the powder, and EDS elemental mapping of (**c**) Al, (**d**) Ti, (**e**) P, (**f**) O, and (**g**) Si elements.

**Figure 4 nanomaterials-12-01158-f004:**
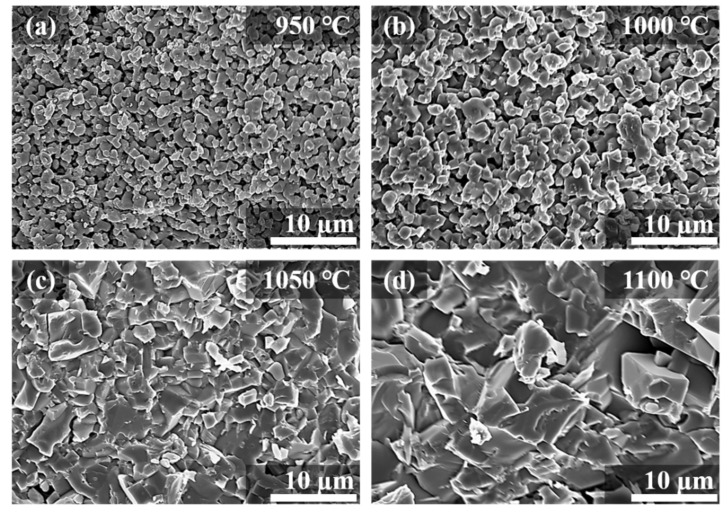
FE-SEM images of LASTP-X pellet sintered at various temperatures (**a**) 950 °C, (**b**) 1000 °C, (**c**) 1050 °C, and (**d**) 1100 °C.

**Figure 5 nanomaterials-12-01158-f005:**
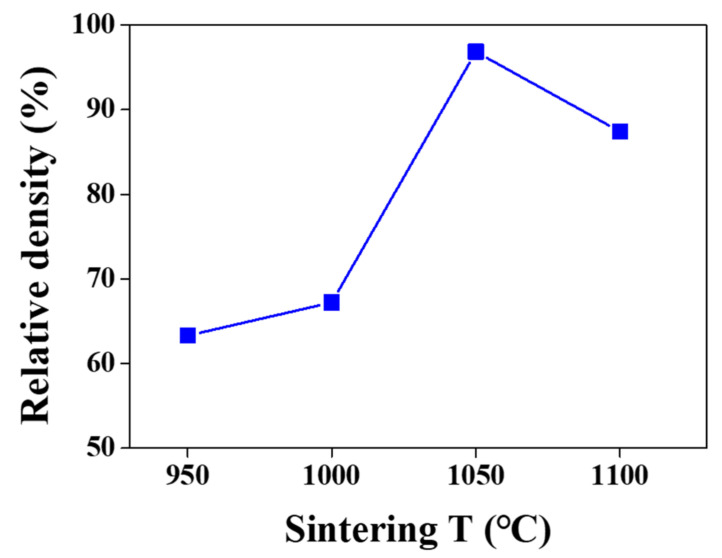
Relative densities of pellets at various sintering temperatures.

**Figure 6 nanomaterials-12-01158-f006:**
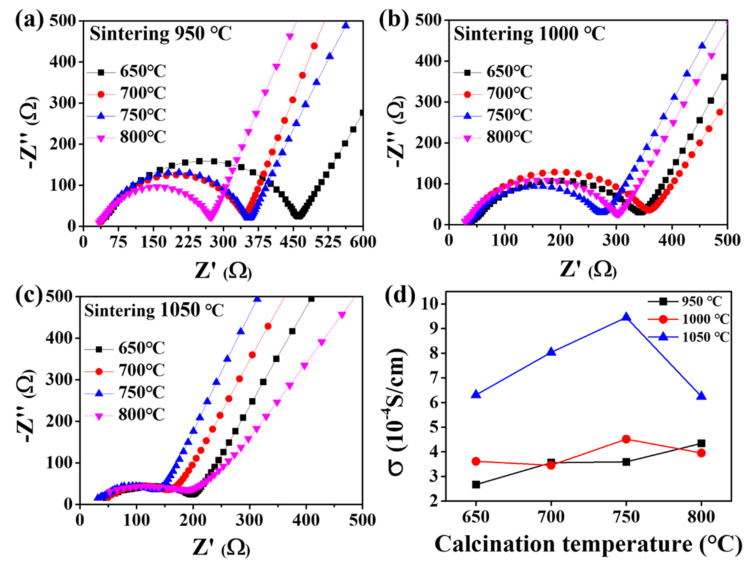
Impedance spectra for LASTP-750 pellet sintered at (**a**) 950 °C, (**b**) 1000 °C, (**c**) 1050 °C, and (**d**) temperature dependence of LASTP-750 ionic conductivity.

**Figure 7 nanomaterials-12-01158-f007:**
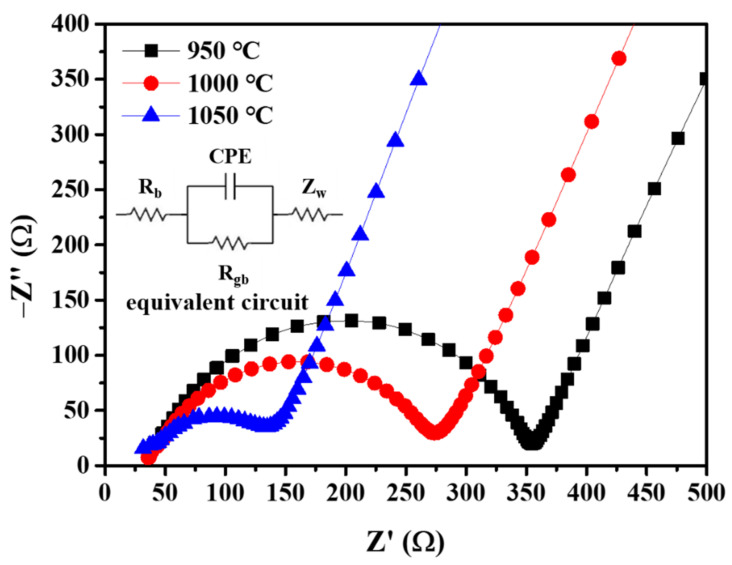
Impedance spectra for LASTP-750 pellet.

**Figure 8 nanomaterials-12-01158-f008:**
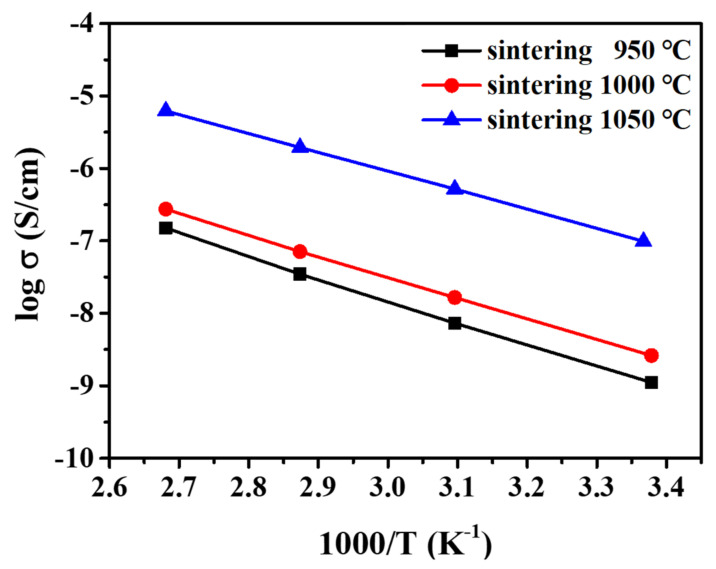
Arrhenius plots of LASTP-750 pellets sintered at various temperatures.

**Table 1 nanomaterials-12-01158-t001:** Relative densities of the LASTP pellets sintered at various temperatures.

Sintering *T* (°C)	Relative Density (%)
950	63.30
1000	67.23
1050	96.84
1100	87.43 (fracture)

**Table 2 nanomaterials-12-01158-t002:** Thicknesses and ionic conductivities of pellets sintered at various temperatures.

	Sintering	950 °C		1000 °C		1050 °C	
Calcination		*t* (cm)	*σ* (10^−4^ S cm^−1^)	*t* (cm)	*σ* (10^−4^ S cm^−1^)	*t* (cm)	*σ* (10^−4^ S cm^−1^)
650 °C		0.097	2.670	0.097	3.615	0.098	6.390
700 °C		0.096	3.558	0.097	3.453	0.097	8.036
750 °C		0.098	3.591	0.100	4.514	0.100	9.455
800 °C		0.094	4.344	0.093	3.952	0.093	6.244

**Table 3 nanomaterials-12-01158-t003:** Bulk and grain boundary conductivities of the LASTP pellets.

Sintering *T* (°C)	*σ_b_* (mS cm^−1^)	*σ_gb_* (mS cm^−1^)
950	2.8967	0.3966
1000	3.2524	0.5350
1050	3.6651	1.2355

**Table 4 nanomaterials-12-01158-t004:** The activation energies and ionic conductivities of the LASTP pellets sintered at various temperatures.

Sintering *T* (°C)	*E_a_* (*σ_T_*) (eV)	*σ_T_* (10^−4^ S cm^-1^)
950	0.263	3.591
1000	0.249	4.514
1050	0.226	9.455

## Data Availability

The data presented in this study are available on request from the corresponding author. The data are not publicly available as it is part of the ongoing project.

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
