# Peer review of "Synthesis of Superionic Conductive Li1+x+yAlxSiyTi2−xP3−yO12 Solid Electrolytes"

_nanomaterials, 2022, doi:10.3390/nano12071158_

Round 1
Reviewer 1 Report
The subject of the paper is very actual and important.
It is well writen and didactic,
Author Response
Response to reviewer 1
Manuscript ID: nanomaterials-1648639
General comment: The subject of the paper is very actual and important. It is well written and didactic.
Response: We thank the reviewer for appreciating the work. We are grateful to the reviewer for his positive recommendation and for accepting the work in its original form.
Reviewer 2 Report
The manuscript titled “Synthesis of superionic conductive Li1+x+yAlxSiyTi2-xP3-yO12 solid electrolytes” reports the preparation of sodium superionic conductor-type Li1+x+yAlxSiyTi2-xP3-yO12 (LASTP) solid electrolyte. In my opinion, the authors should make some changes and clarify a few points in the manuscript before it can be accepted for publication. The details are as follows.
- The authors should give a table to compare the property/performance of the as-obtained LASTP with other typical solid electrolytes that have been reported in the literature.
- How about the performance of LASTP in the half cell or full cell? The author should demonstrate it and give experimental evidence.
- How about the stability of the LASTP? XRD, SEM, and TEM of LASTP after tests should be provided.
- How will the porosity and surface area affect the property/performance of the as-obtained LASTP?
- Some related references should be cited, such as 10.1016/j.cclet.2021.03.032, 10.1002/adma.201601925, 10.1007/s40820-021-00780-7.
Reviewer 3 Report
The article Synthesis of superionic conductive Li1+x+yAlxSiyTi2-xP3-yO12 solid electrolytes is devoted to the study of methods for obtaining new solid electrolytes for sodium superionic conductors, which have great potential for use as solid-state batteries. Undoubtedly, the results presented by the authors are of high scientific novelty and practical significance, and are also promising for practical research. In general, the presented results of the study can be accepted for publication after the authors provide answers to all the questions raised by the reviewer during the reading of the article.
1. In the abstract, the authors need to more clearly state the purpose and relevance of this work.
2. The authors should explain the choice of annealing conditions, as well as how exactly the samples were cooled down, together with the furnace or immediately after annealing, the samples were placed in air.
3. The authors should explain exactly how the density of the samples was calculated.
4. The authors should present the results of the size diagrams of the synthesized powders.
5. The authors should present the results of the phase composition.
6. Conclusion requires significant revision.
Round 2
Reviewer 3 Report
The authors answered all the questions of the reviewer, the article can be accepted for publication.